# Fouling Mechanisms Analysis via Combined Fouling Models for Surface Water Ultrafiltration Process

**DOI:** 10.3390/membranes10070149

**Published:** 2020-07-10

**Authors:** Bin Huang, Hangkun Gu, Kang Xiao, Fangshu Qu, Huarong Yu, Chunhai Wei

**Affiliations:** 1Department of Municipal Engineering, School of Civil Engineering, Guangzhou University, Guangzhou 510006, China; huangbinstudy@163.com (B.H.); ghk455818213@163.com (H.G.); qufs@gzhu.edu.cn (F.Q.); huarongyu@gmail.com (H.Y.); 2Key Laboratory for Water Quality and Conservation of the Pearl River Delta, Ministry of Education, Guangzhou 510006, China; 3College of Resources and Environment, University of Chinese Academy of Sciences, Beijing 100049, China; kxiao@ucas.ac.cn

**Keywords:** ultrafiltration, membrane fouling, fouling model, surface water, coagulation, backwash

## Abstract

Membrane fouling is still the bottleneck affecting the technical and economic performance of the ultrafiltration (UF) process for the surface water treatment. It is very important to accurately understand fouling mechanisms to effectively prevent and control UF fouling. The rejection performance and fouling mechanisms of the UF membrane for raw and coagulated surface water treatment were investigated under the cycle operation of constant-pressure dead-end filtration and backwash. There was no significant difference in the UF permeate quality of raw and coagulated surface water. Coagulation mainly removed substances causing turbidity in raw surface water (including most suspended particles and a few organic colloids) and thus mitigated UF fouling effectively. Backwash showed limited fouling removal. For the UF process of both raw and coagulated surface water, the fittings using single models showed good linearity for multiple models mainly due to statistical illusions, while the fittings using combined models showed that only the combined complete blocking and cake layer model fitted well. The quantitative calculations showed that complete blocking was the main reason causing flux decline. Membrane fouling mechanism analysis based on combined models could provide theoretical supports to prevent and control UF fouling for surface water treatment.

## 1. Introduction

The ultrafiltration (UF) membrane with a nominal pore size of 10–20 nm can reject suspended particles completely and colloids, bacteria and viruses efficiently, while keeping higher permeability than a tight UF membrane with a nominal pore size of a few nanometers, making it a promising advanced technology for drinking water production from conventional surface water resources (e.g., river, lake and reservoir). The rejection capability of UF membrane is generally not affected by the feed quality (e.g., turbidity) and operational conditions (e.g., pressure or flux), thus resulting in very stable and excellent permeate quality. UF membrane modules have also some advantages including the modular design and assembly, compact structure, small footprint and automatic operation. Therefore, the large-scale (up to 0.6 million m^3^/d) UF systems have been gradually implemented in conventional surface water treatment plants in recent years and more applications would be expected in the future based on the increasing demand for high quality drinking water and the decreasing cost of UF technology [1]. However, membrane fouling is inevitably developed with the filtration time, which derives from the deposition on membrane surface and/or blocking membrane pores by suspended particles, colloids and microorganisms. Membrane fouling would increase the operational pressure under the constant-flux mode or decrease the membrane flux under the constant-pressure mode, increasing the maintenance cost of the UF process [2]. Therefore, it is of great significance to accurately understand membrane fouling mechanisms for the effective prevention and control of UF membrane fouling.

Several mathematical models have been developed to describe the mechanisms of pore blocking and cake layer fouling caused by the presence of contaminants during the filtration process [3]. Hermans and Bredée [4] initially proposed four classical single models (cake layer, intermediate blocking, standard blocking and complete blocking) for constant-pressure dead-end filtration based on the filter cloth tests. Then Grace [5] proposed the common differential equation, which unified the above-mentioned four classical single models via adjusting the values of two constants. Hermia [6] further improved the physical basis of the intermediate blocking model and deduced the linear expressions of four classical single models (shown in Table 1). Bowen et al. [7] and Cho et al. [8] subsequently introduced the single models into the field of microfiltration (MF) and UF. With the development of research on fouling models, Ho et al. [9] proposed a combined model describing the initial membrane pore blocking and the later cake layer. Furthermore, Bolton et al. [10] developed five combined models (complete blocking and cake layer, intermediate blocking and cake layer, standard blocking and cake layer, complete blocking and standard blocking, intermediate blocking and standard blocking) via combining two single models (shown in Table 1 for specific formula). The development of the above-mentioned models was mostly based on protein solution filtration tests.

The single models have been gradually applied in the water treatment field using MF/UF [11,12,13,14]. Schippers and Verdouw firstly proposed a modified fouling index (MFI) based on the cake layer model using 0.45 µm MF membrane to characterize the particulate fouling potential of water samples [15], which became the theoretical basis of the standard methods for MFI measurement [16]. Jin et al. [17] conducted MFI measurements twice to eliminate the effects of membrane pore blocking and proposed the cake fouling index. Due to the ubiquitous colloids in natural water bodies (e.g., river, lake and sea), Boerlage et al. [18] employed UF membranes to develop the MFI-UF measurement to cover colloidal effects on fouling potential. Sim et al. [19] further proposed the cross-flow sampling MFI-UF measurement to cover the crossflow effects on fouling potential. However, there is little information available in the literature about the application of combined models for the UF membrane fouling analysis in the surface water treatment [20,21].

This study aimed to clarify UF membrane fouling mechanisms during both real raw and coagulated surface water filtration via mathematical model fitting including the above-mentioned single and combined fouling models, and investigate the performance and mechanisms of the coagulation pretreatment and backwash for UF membrane fouling control. The findings from this study would provide theoretical supports for the prevention and control of UF membrane fouling.

## 2. Materials and Methods

### 2.1. Raw Surface Water

The raw surface water was sampled from the landscape lake in the university town campus of Guangzhou University. The turbidity and dissolved organic carbon (DOC) of raw surface water were 33.8 NTU and 20.1 mg/L, respectively.

### 2.2. Coagulation Pretreatment

In this study, a coagulation device (model ZR4–6, Zhongrun, Shenzhen, China) was used to conduct the coagulation pretreatment of raw surface water. FeCl_3_ was selected as the coagulant. The coagulation sequence was coagulant spiking → rapid stirring for 30 s at 500 r/min → slow stirring for 300 s at 150 r/min → slow stirring for 600 s at 100 r/min → sedimentation for 15 min. The pH of raw surface water was 7.34 and no pH control was done for coagulation experiments. Turbidity removal under different FeCl_3_ dosage (1–20 mg/L) was firstly investigated. The turbidity of coagulated surface water showed a rapid decrease followed by a steady trend with the increase of FeCl_3_ dosage. The inflection point of the curve of turbidity vs. FeCl_3_ dosage was around 10 mg/L, which was selected as the optimum dosage considering turbidity removal and coagulant cost. Then, sufficient coagulated surface water samples were prepared under the optimum dosage for subsequent UF experiments. The turbidity and DOC of the coagulated surface water were 3.39 NTU and 17.5 mg/L, which were reduced by 90% and 13% compared with the raw surface water, respectively. This indicated that coagulation mainly removed suspended particles (i.e., the main turbidity substances) and a small part of organic colloids (characterized as DOC) in this study.

### 2.3. UF Experiments

A laboratory-scale constant-pressure dead-end filtration system (shown in Figure 1) was used for UF experiments in this study. Compressed nitrogen was used to pressurize the water sample in the stainless steel influent tank with an effective volume of 10 L into the filtration cell (Amicon 8400, Millipore, Burlington, MA, USA) for constant-pressure dead-end filtration. A flat-sheet UF membrane with a molecular weight cut-off of 150 kDa and material of polyvinylidene fluoride (Koch, Wilmington, NC, USA) was used. The mass of UF permeate was weighed by an electronic balance (ME4002E, Mettler Toledo, Greifensee, Switzerland) and sent to the computer for real time recording. Based on the measured temperature of the UF permeate, the density was determined, and thus the mass was further converted into volume. The instantaneous filtration rate was obtained by the numerical differentiation between the UF permeate volume and filtration time, and thus the instantaneous membrane flux calculation and membrane fouling model fitting were carried out based on the effective UF membrane area.

Before filtering surface water samples (raw surface water or coagulated surface water), pure water was filtered for 10 min under 100 kPa to record the pure water flux of the new membrane for pure membrane resistance calculation. Then, surface water was filtered for 1 h under 100 kPa. After filtration, the UF membrane was reversely placed in the filtration cell and backwashed with 5% of UF permeate under 150 kPa. After backwash, the UF membrane was placed in the normal direction and pure water was filtered for 10 min under 100 kPa again to record the membrane flux for residual resistance calculation after backwash. Thus, a complete cycle of filtration followed by backwash (total water yield 95%) was finished. A total of 5 cycles were conducted to simulate the real UF process for surface water treatment.

### 2.4. Water Quality Analysis

Turbidity of all water samples was directly measured by a portable turbidity meter (WGZ-4000B, Xinrui, Shanghai, China). Raw and coagulated surface water samples were pre-filtered through a 0.45 μm syringe filter to determine DOC by an organic carbon analyzer (TOC-L, Shimadzu, Kyoto, Japan). UF permeate samples were directly measured for DOC.

### 2.5. Membrane Fouling Mechanism Analysis

The single and combined fouling models for the constant-pressure UF process [4,6,10] was shown in Table 1. Origin 2018 software was used to fit the UF data of raw and coagulated surface water to the model equations in order to clarify the fouling mechanisms in this study. For single and combined membrane fouling models, linear and nonlinear fitting were performed respectively. The coefficient of determination R^2^ (i.e., the ratio of sum of squares for regression to the sum of squares for total, SSR/SST, with a value range of 0–1) characterizes the quality of the fitting results. On the basis of passing the parameter *t* test (<0.05), R^2^ > 0.95 can be generally considered as a successful fitting, the closer to 1, the better the fitting.

## 3. Results and Discussion

### 3.1. Rejection Performance by UF for the Surface Water Treatment

The turbidity and DOC of UF permeate for raw and coagulated surface water (shown in Figure 2) were 0.38–0.56 NTU and 17.5–20.1 mg/L, 0.32–0.36 NTU and 16.9–17.5 mg/L, respectively, showing slightly better UF permeate quality for the coagulated than raw surface water. This was mainly due to that most of the substances removed by coagulation (suspended particles and a small part of organic colloids) could be directly rejected by the UF membrane in this study, demonstrating the stable rejection by the UF membrane. The rejection of turbidity and DOC by the UF membrane for raw and coagulated surface water was 98.3–98.9% and 85.3–90.6%, 1.1–8.4% and 1.8–3.8%, respectively, indicating that UF membrane achieved high turbidity rejection and low DOC rejection. Turbidity of surface water was generally composed of suspended particles and organic colloids (component of DOC). Based on the high turbidity rejection and low DOC rejection by UF membrane in this study, it could be preliminarily inferred that suspended particles were the main membrane foulants from the perspective of UF rejection.

### 3.2. Membrane Fouling of UF for the Surface Water Treatment

The UF flux of the coagulated surface water during the first to fifth filtration cycle was significantly higher than that of raw surface water during the corresponding filtration cycle (shown in Figure 3a–b), indicating the significant UF fouling mitigation performance by coagulation. As a classical electrolyte coagulant, FeCl_3_ used in this study can firstly neutralize the negatively charged colloids, then enhance these colloids aggregated into small particles, further making small particles aggregated into big particles via adsorption bridging, and finally enhance big particles settling from water. Thus, FeCl_3_ coagulation could change the content and size of suspended particles and colloids in surface water [22,23], resulting in a 90% decrease of turbidity (mainly suspended particles) and 13% decrease of organic colloids (measured by DOC) after coagulation in this study. This further significantly reduced the fouling potential of coagulated surface water and thus UF membrane fouling. The initial UF flux (837 L/m^2^/h) of coagulated surface water during the first filtration cycle was slightly lower than the corresponding pure water flux (853 L/m^2^/h), which was derived from the simultaneous occurrence of fouling during the pressure regulation process (about 1 min) before the filtration test. The initial UF flux (504 L/m^2^/h) of the raw surface water during the first filtration cycle was significantly lower than the corresponding pure water flux (772 L/m^2^/h), which derived from the heavy fouling caused by the high turbidity of raw surface water (about 10 times of coagulated surface water) during the pressure regulation process before the filtration test. Park et al. also found that the higher the influent turbidity, the faster the membrane flux decreased [24]. Resistance distribution at the end of each filtration (shown in Figure 3c–d) indicated that the removed resistance by backwash accounted for 56.9–67.7% (average 60.6%) and 38.3–59.5% (average 50.1%) of the total fouling resistance developed during the UF process for the raw and coagulated surface water, respectively. The backwash performance was slightly better for the raw surface water than the coagulated surface water, which was mainly due to the higher fouling resistance for the raw surface water than the coagulated surface water. However, the residual resistance after backwash showed a gradual increase with the filtration cycle for both the raw and coagulated surface water, indicating the limited performance for fouling removal by the simple backwash used in this study. This was similar to the findings from Jang et al. [25] that only backwash was less effective than the combined back and forward wash for UF membrane fouling control.

### 3.3. Analysis of UF Membrane Fouling Mechanisms Based on Single Models

As a typical example, the fitting analysis of the second UF test for raw and coagulated surface water using single models was shown in Figure 4. The results of the other four UF tests were the same. All four single fouling models (cake layer, standard blocking, intermediate blocking and complete blocking) during the UF process of raw surface water showed good linear fitting (R^2^ > 0.96, *t* < 0.01). Three single models (cake layer, standard blocking and intermediate blocking) during the UF process of the coagulated surface water also showed good linear fitting (R^2^ > 0.96, *t* < 0.01). From a statistical point of view, this indicated that multiple fouling mechanisms occurred at the same time. The substances in the surface water had generally a wide size distribution (1 nm to 1 mm). The components significantly larger than the UF membrane pore size (mainly suspended particles and some large-size colloids) could form cake layer fouling, the components equivalent to the UF membrane pore size (mainly colloids) could form complete blocking and intermediate blocking fouling, and the components significantly smaller than the UF membrane pore size (mainly soluble substances and some small colloids) could form standard blocking fouling. Therefore, the four fouling mechanisms in the UF process of the surface water could occur simultaneously in theory. Wei and Amy [26] found the simultaneous occurrence of two fouling mechanisms during the UF process of the wastewater treatment plant effluent. Corbaton et al. [27] found that single models did not characterize the membrane fouling mechanism. Li et al. [28] found multiple fouling mechanisms involved in the UF process of river water. Once the simultaneous occurrence of multiple fouling mechanisms, the single models may produce statistical illusions for the UF membrane fouling mechanism analysis, especially for the quantitative evaluation of the contribution of a single fouling mechanism. Thus, the applicability of single models should be further verified by the combined models.

### 3.4. Analysis of UF Membrane Fouling Mechanisms Based on Combined Models

Nonlinear fitting between the permeate volume *V* and filtration time *T* using combined models was conducted for the above-mentioned UF data (shown in Table 2). The combined standard blocking and cake layer model did not converge. The combined models of complete blocking and standard blocking, intermediate blocking and standard blocking and intermediate blocking and cake layer did not pass the parameter *t* test. Only the combined complete blocking and cake layer model fitted well (R^2^ was 0.9935 and 0.9948 for the UF process of the raw and coagulated surface water, respectively; *t* < 0.01, shown in Figure 5). Among the linear fitting results of four single models (shown in Table 2), the linearity of the complete blocking model for the UF process of the raw and coagulated surface water was the worst, while the linearity of the cake layer model and the standard blocking model was the best. Thus, it could be intuitively speculated that the combined standard blocking and cake layer model fitted the best among the combined models. However, the best-fitting combined model was the combined complete blocking and cake layer model in fact, indicating that single models might not be applicable when multiple fouling mechanisms occurred simultaneously. Due to the existence of colloids equivalent to UF membrane pore size and suspended particles much larger than the UF membrane pore size in surface water, the fouling mechanisms of complete blocking and the cake layer could occur simultaneously during the UF process of the surface water. Li et al. [28] also found the simultaneous occurrence of standard blocking (or intermediate blocking) and cake layer fouling during the UF process of flocculated but unsettled river water. Xing et al. [29] employed the hybrid adsorption/oxidation and the UF process for the algae-laden surface water treatment and found the simultaneous occurrence of multiple fouling mechanisms.

Since the combined model derives from two single models, the fitting parameters can be used to quantitatively evaluate the contribution of a single model to the combined model. According to the definition of fouling models, cake layer fouling increases the filtration resistance, resulting in a flux decline of Δ*J*/*J*_0_ = *K_c_**J*_0_*V*/(1 + *K_c_**J*_0_*V*) ≈ *K_c_J*_0_*V* (when *V* is small). When complete blocking fouling occurs, the blocked membrane pores lose filtration capacity, resulting in a flux decline of Δ*J*/*J*_0_ = (*K_b_*/*J*_0_)*V*. Therefore, the ratio of the above-mentioned two values of Δ*J*/*J*_0_ (i.e., *K_c_J*_0_/(*K_b_*/*J*_0_)) can be used to quantitatively evaluate the individual contribution of the cake layer and complete blocking fouling to the decline in membrane flux. The ratio of *K_c_J*_0_/(*K_b_*/*J*_0_) for the UF process of the raw and coagulated surface water was 0.052 and 0.027 (i.e., the percentage of complete blocking fouling for flux decline was 95.1% and 97.4%), respectively, indicating that complete blocking fouling was the main reason for the UF flux decline in this study. This seemed to be somewhat contradictory to the previous deduction from the analysis of water quality before and after UF that “from the perspective of UF rejection, suspended particles were the main membrane foulants”. The main reasons were the different fouling characteristics of the complete blocking formed by colloids and the cake layer formed by suspended particles as well as the different concentrations of colloids and suspended particles in the surface water. The blocked membrane pores lost the filtration capacity (i.e., the resistance was infinite) when complete blocking fouling occurred, while the resistance caused by cake layer fouling was finite. Therefore, the flux decline caused by complete blocking fouling formed by colloids would be higher than that caused by the cake layer formed by suspended particles with the same amount to colloids. Despite no direct measurement in this study, the concentration of suspended particles could be roughly estimated as 33.8 mg/L and 3.39 mg/L for the raw and coagulated surface water, respectively, according to the turbidity and the conversion factor between turbidity and standard SiO_2_ concentration. The low-concentration suspended particles in surface water in this study were not enough to quickly form a uniform and dense cake layer, resulting in a low flux decline caused by cake layer fouling. Membrane pores were also not effectively covered by the cake layer and thus more colloids with equivalent size to membrane pores in the surface water could reach membrane surface and form complete blocking, resulting in a high flux decline. Bolton et al. [11] investigated the UF process of the bovine serum protein solution with a concentration of up to 2500 mg/L and found *K_c_J*_0_/(*K_b_*/*J*_0_) of 28.3 (i.e., the cake layer and complete blocking fouling accounted for 96.6% and 3.4% of the flux decline, respectively), indicating that the dominant cake layer fouling occurred under high concentration conditions. Li et al. [28] investigated the UF process of flocculated but the unsettled river water and found the simultaneous occurrence of minor standard blocking (or intermediate blocking) and dominant cake layer fouling due to the existence of high-concentration flocs in flocculated but unsettled river water. It should be pointed out that if the property of the surface water and/or UF membrane (e.g., particle/pore size distribution) is changed, the fouling mechanism would also change because it is intrinsically dependent on the interactions between the surface water and UF membrane.

Table 3 lists the fitted characteristic parameter values, R^2^ and *K_c_J*_0_/(*K_b_*/*J*_0_) of the combined complete blocking and cake layer model for the UF process of the raw and coagulated surface water. The *K_c_* and *K_b_* of raw surface water were higher than that of the coagulated surface water, showing a significant correlation with water quality. The concentration of suspended particles and organic colloids (measured as turbidity) in the raw surface water was higher than that of the coagulated surface water. During the UF process of the surface water in this study, suspended particles with a size larger than 0.45 µm could form cake layer fouling, while some organic colloids with the same or close molecular weight cut-off (150 kDa) of the UF membrane could form complete blocking fouling. The *K_c_J*_0_/(*K_b_*/*J*_0_) of the coagulated surface water was lower than that of the raw surface water, meaning that cake layer fouling accounted for a lower contribution to flux decline during the UF process of the coagulated surface water than the raw surface water. This reflected the performance of coagulation pretreatment to mainly remove suspended particles. In addition, the *K_c_J*_0_/(*K_b_*/*J*_0_) (i.e., the contribution of cake layer fouling) of both the raw and coagulated surface water showed an upward trend with increasing filtration cycles, which might be related to foulants accumulation caused by the limited backwash performance.

## 4. Conclusions

This study investigated the rejection performance, membrane flux changes and membrane fouling mechanisms of the constant-pressure UF process of the raw and coagulated surface water. The average rejection of turbidity and DOC by the UF membrane for the raw and coagulated surface water was 98.6% and 89.1%, 3.8% and 3.0%, respectively. There was no significant difference in the UF permeate quality of the raw and coagulated surface water under the short-term filtration conditions in this study. Coagulation mainly removed turbidity substances (including most suspended particles and a small part of organic colloids) in the raw surface water, thereby significantly reducing UF membrane fouling. Simple backwash showed limited performance to remove membrane fouling. Linear fitting of single membrane fouling models to UF data showed good linearity (R^2^ > 0.96) of four models (cake layer, standard blocking, intermediate blocking and complete blocking) and three models (cake layer, standard blocking and intermediate blocking) for the raw and coagulated surface water, respectively, indicating statistically the simultaneous occurrence of multiple fouling mechanisms. Non-linear fitting of combined membrane fouling models showed firstly that only the combined complete blocking and cake layer model fitted well (R^2^ > 0.99), indicating that the single membrane fouling models produced statistical illusions and thus could not truly describe the UF process of the real surface water with the simultaneous occurrence of multiple fouling mechanisms. The quantitative calculation from the combined model showed for the first time that the complete blocking fouling formed by the colloids was the main reason for the decline of UF membrane flux (accounting for more than 95%). This study employed real raw and coagulated surface water and simulated the representative operation mode of the filtration-backwash cycle in the UF plant for the surface water treatment. The findings from this study would provide theoretical supports for the mechanism analysis and the control method of UF membrane fouling in surface water treatment plants.

## Figures and Tables

**Figure 1 membranes-10-00149-f001:**
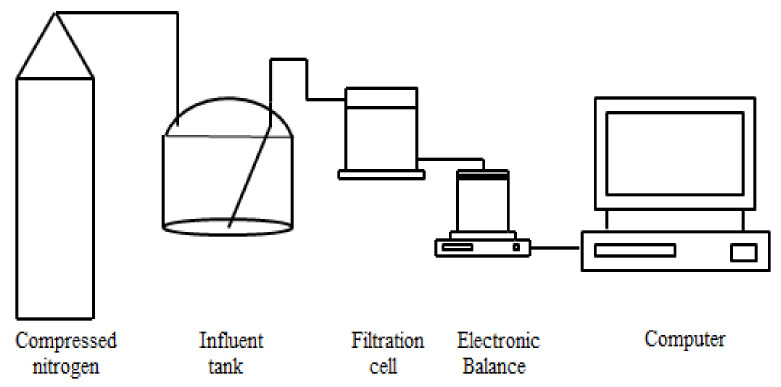
Constant-pressure dead-end filtration system.

**Figure 2 membranes-10-00149-f002:**
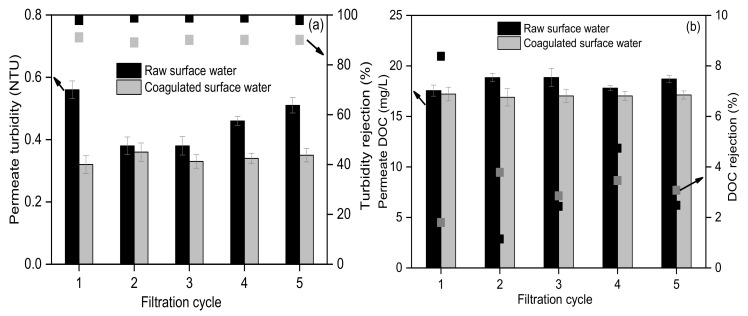
Rejection performance of turbidity (**a**) and dissolved organic carbon (DOC) (**b**) by UF membrane for the raw and coagulated surface water.

**Figure 3 membranes-10-00149-f003:**
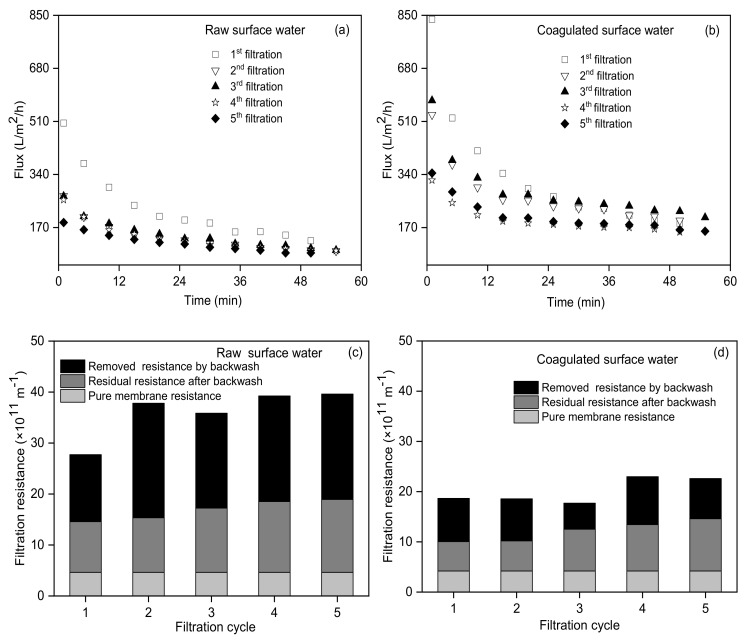
UF membrane fouling including a flux decline for the raw surface water (**a**) and coagulated surface water (**b**), and resistance distribution at the end of filtration for the raw surface water (**c**) and coagulated surface water (**d**).

**Figure 4 membranes-10-00149-f004:**
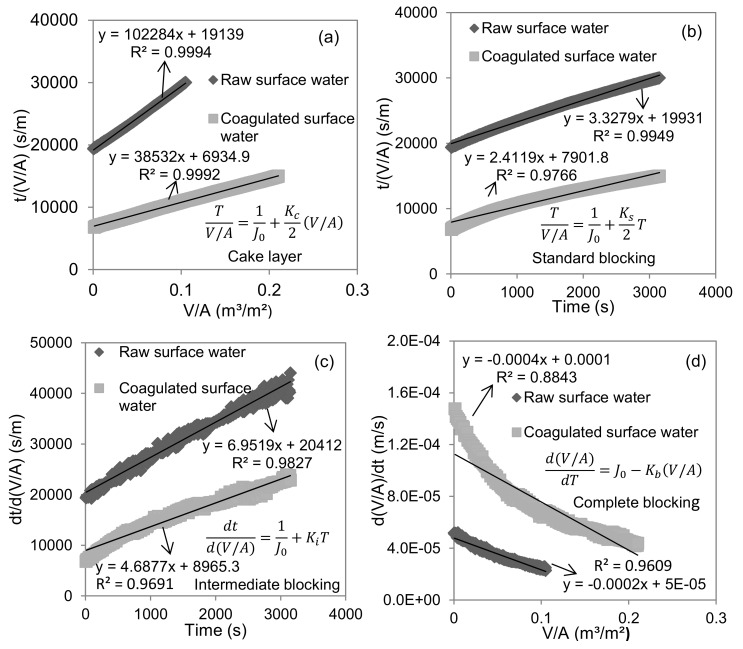
Linear fitting of single models for the UF membrane fouling mechanisms analysis in terms of cake layer model fitting (**a**), standard blocking model fitting (**b**), intermediate blocking model fitting (**c**) and complete blocking model fitting (**d**).

**Figure 5 membranes-10-00149-f005:**
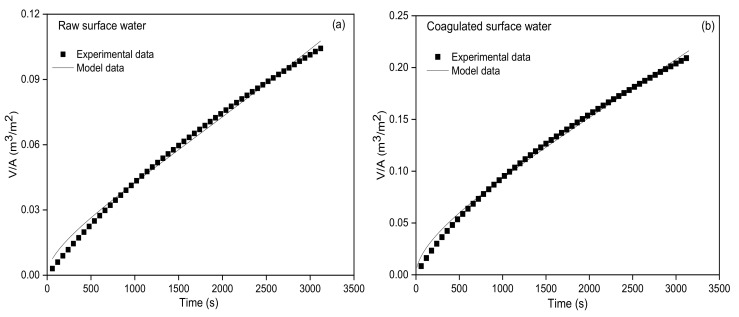
Graphic fitting of the combined complete blocking and cake layer model for the raw surface water (**a**) and coagulated surface water (**b**).

**Table 1 membranes-10-00149-t001:** Membrane fouling models for the constant-pressure ultrafiltration (UF) process.

Model	Equation *	Characteristic Parameters	Schematic Diagram	References
Cake layer	$\frac{T}{\left(V/A\right)}=\frac{1}{J_{0}}+\frac{K_{c}}{2}\left(V/A\right)$	*K_c_* (s/m^2^)	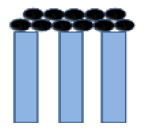	[4,6]
Complete blocking	$\frac{d\left(V/A\right)}{dT}=J_{0}-K_{b}\left(V/A\right)$	*K_b_* (s^−1^)	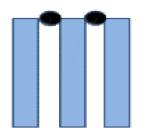	[4,6]
Intermediate blocking	$\frac{dT}{d\left(V/A\right)}=\frac{1}{J_{0}}+K_{i}T$	*K_i_* (m^−1^)	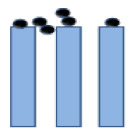	[4,6]
Standard blocking	$\frac{T}{\left(V/A\right)}=\frac{1}{J_{0}}+\frac{K_{s}}{2}T$	*K_s_* (m^−1^)	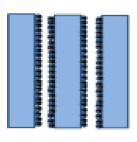	[4,6]
Complete blocking and Cake layer	$\frac{V}{A}=\frac{J_{0}}{K_{b}}\left(1-exp\left(\frac{-K_{b}}{K_{c}J_{0}^{2}}\left(\sqrt{1+2K_{c}J_{0}^{2}T}-1\right)\right)\right)$	*K_c_* (s/m^2^), *K_b_* (s^−1^)	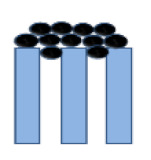	[10]
Intermediate blocking and Cake layer	$\frac{V}{A}=\frac{1}{K_{i}}ln\left(1+\frac{K_{i}}{K_{c}J_{0}}\left(\sqrt{1+2K_{c}J_{0}^{2}T}-1\right)\right)$	*K_c_* (s/m^2^), *K_i_* (m^−1^)	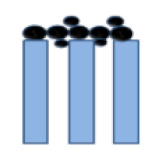	[10]
Complete blocking and Standard blocking	$\frac{V}{A}=\frac{J_{0}}{K_{b}}\left(1-exp\left(\frac{-2K_{b}T}{2+K_{s}J_{0}T}\right)\right)$	*K_b_* (s^−1^), *K_s_* (m^−1^)	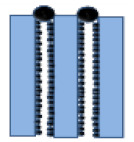	[10]
Intermediate blocking and Standard blocking	$\frac{V}{A}=\frac{1}{K_{i}}ln\left(1+\frac{2K_{i}J_{0}T}{2+K_{s}J_{0}T}\right)$	*K_i_* (m^−1^), *K_s_* (m^−1^)	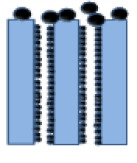	[10]
Standard blocking and Cake layer	$T=\frac{1}{K_{s}\left(\frac{V}{A}\right)-2}\left(\frac{K_{s}K_{c}{\left(\frac{V}{A}\right)}^{3}}{2}-K_{c}{\left(\frac{V}{A}\right)}^{2}-\frac{2\left(\frac{V}{A}\right)}{J_{0}}\right)$	*K_c_* (s/m^2^), *K_s_* (m^−1^)	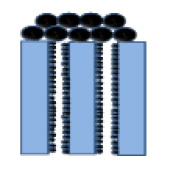	[10]

* *T*—filtration time (s); *V*—permeate volume (m^3^); *A*—membrane area (m^2^); *J*_0_—initial membrane flux (m/s).

**Table 2 membranes-10-00149-t002:** Fitting results of combined and single models for the UF process of the surface water.

Model	Raw Surface Water		Coagulated Surface Water	
	Non-Linear/Linear Fitting R^2^	Characteristic Parameters	Non-Linear/Linear Fitting R^2^	Characteristic Parameters
Complete blocking and Cake layer	0.9935	*K_b_* = 1.12 s^−1^ *K_c_* = 1.15 × 10^6^ s/m^2^	0.9948	*K_b_* = 0.35 s^−1^ *K_c_* = 1.94 × 10^5^ s/m^2^
Intermediate blocking and Cake layer	0.9423	*K_i_* = 2.17 × 10^−6^ m^−1^ * *K_c_* = 5.90 × 10^5^ s/m^2^	0.9908	*K_i_* = 2.83 × 10^−7^ m^−1^* *K_c_* = 1.08 × 10^5^ s/m^2^
Standard blocking and Cake layer	Fitting failed due to no convergence		Fitting failed due to no convergence	
Complete blocking and Standard blocking	0.6428	*K_b_* = 1.64 × 10^−5^ s^−1^ * *K_s_* = 22.90 m^−1^ *	0.9102	*K_b_* = 2.51 × 10^−6^ s^−1^ * *K_s_* = 28.15 m^−1^ *
Intermediate blocking and Standard blocking	0.7989	*K_i_* = 39.77 m^−1^ *K_s_* = 1.77 m^−1^ *	0.9503	*K_i_* = 11.65 m^−1^ *K_s_* = 0.84 m^−1^ *
Cake layer	0.9994	*K_c_* = 5.1 × 10^4^ s/m^2^	0.9992	*K_c_* = 1.93 × 10^4^ s/m^2^
Intermediate blocking	0.9827	*K_i_* = 6.95 m^−1^	0.9691	*K_i_* = 4.69 m^−1^
Complete blocking	0.9609	*K_b_* = 2 × 10^−4^ s^−1^	0.8843	*K_b_* = 4 × 10^−4^ s^−1^
Standard blocking	0.9949	*K_s_* = 1.66 m^−1^	0.9766	*K_s_* = 1.21 m^−1^

* Failed the *t* test at the 0.05 significance level of the characteristic parameter, i.e., *t* > 0.05.

**Table 3 membranes-10-00149-t003:** Fitting results of the combined complete blocking and cake layer model for the UF process of the surface water.

Filtration Cycle	Raw Surface Water			Coagulated Surface Water		
	Characteristic Parameter	R^2^	*K_c_J*_0_/(*K_b_*/*J*_0_)	Characteristic Parameter	R^2^	*K_c_J*_0_/(*K_b_*/*J*_0_)
1st	*K_b_* = 0.27 s^−1^ *K_c_* = 1.78 × 10^5^ s/m^2^	0.9931	0.033	*K_b_* = 0.24 s^−1^ *K_c_* = 9.70 × 10^4^ s/m^2^	0.9938	0.021
2nd	*K_b_* = 1.12 s^−1^ *K_c_* = 1.15 × 10^6^ s/m^2^	0.9945	0.052	K_b_ = 0.35 s^−1^ *K_c_* = 1.94 × 10^5^ s/m^2^	0.9952	0.027
3rd	*K_b_* = 0.61 s^−1^ *K_c_* = 5.55 × 10^5^ s/m^2^	0.9949	0.046	*K_b_* = 0.29 s^−1^ *K_c_* = 1.55 × 10^5^ s/m^2^	0.9954	0.027
4th	*K_b_* = 0.71 s^−1^ *K_c_* = 6.83 × 10^5^ s/m^2^	0.9948	0.049	*K_b_* = 0.56 s^−1^ *K_c_* = 4.10 × 10^5^ s/m^2^	0.9953	0.037
5th	*K_b_* = 0.65 s^−1^ *K_c_* = 6.37 × 10^5^ s/m^2^	0.9951	0.049	*K_b_* = 0.82 s^−1^ *K_c_* = 4.93 × 10^5^ s/m^2^	0.9951	0.031

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
