# Peer review of "Fouling Mechanisms Analysis via Combined Fouling Models for Surface Water Ultrafiltration Process"

_membranes, 2020, doi:10.3390/membranes10070149_

Round 1

Reviewer 1 Report

Table 1, add the references of the models.

Line 110, grammar error, (was placed)

Line 171, change filtraiton to filtration.

Line 159, change smal to small

Line 228, change cominbed to combined

Author Response

Comment 1:Table 1, add the references of the models.

Re: Thanks for the comment. We added the references of the models into Table 1 in the revised manuscript.

Comment 2:Line 110, grammar error, (was placed)

Re: Sorry for this grammar mistake. We corrected it in Line 114 in the revised manuscript.

Comment 3:Line 171, change filtraiton to filtration.

Re: Sorry for this editing mistake. We corrected it in Line 171 in the revised manuscript.

Comment 4:Line 159, change smal to small

Re: Sorry for this editing mistake. We corrected it in Line 159 in the revised manuscript.

Comment 5:Line 228, change cominbed to combined

Re: Sorry for this editing mistake. We corrected it in Line 225 in the revised manuscript.

Reviewer 2 Report

This paper is interesting and in my opinion has an importance in this field. The manuscripts has scientific merits since it contains detailed information on the mechanism of the fouling mechanism on UF membrane. I think this paper will be useful for beginners in this field. So I recommend accepting the paper with minor english check, there are numereous mistakes throughout the text. After minor correction of typos this paper is acceptable

Author Response

Comment:This paper is interesting and in my opinion has an importance in this field. The manuscripts has scientific merits since it contains detailed information on the mechanism of the fouling mechanism on UF membrane. I think this paper will be useful for beginners in this field. So I recommend accepting the paper with minor english check, there are numereous mistakes throughout the text. After minor correction of typos this paper is acceptable.

Re: Thanks for your approval. Following the comments of you and other reviewers, we carefully corrected all editing mistakes in red in the revised manuscript.

Reviewer 3 Report

The research was focus on the fouling mechanism in ultrafiltration membrane using a surface water as a feed. The work is interesting, as they demonstrate the mechanism of fouling during ultrafiltration. However, the current form of the manuscript cannot be accepted, the following are my major concerns:

  1. In abstract, please remove the words “Background”; “Methods”, “Results”, and “Conclusions”
  2. What is the difference of this work from the other reported literature, like Water research, 41(8), 1713-1722; Journal of Membrane Science 281, no. 1-2 (2006): 716-725, Journal of Environmental Sciences, 18(5), 880-884.
  3. In introduction, it is important to mention how ultrafiltration used for production of drinking water. Is it a pretreatment for reverse osmosis or distillation process to obtain drinkable water? How it is a help in the overall process for production of clean water?
  4. The surface water was gathered in the lake, does the turbidity and dissolved organic carbon is similar if gathered in a different day?
  5. Is the fouling mechanism is similar to the authors finding, if the property of the surface water changed?
  6. If the MWCO of the UF membrane was changed, is the fouling mechanism is similar? Why the author chose PVDF UF membrane with MWCO of 150 kDa? MWCO of UF membrane had a large range. It is interesting if the author could include this in their study.
  7. In figure 2, the author mentioned that there was no significant change on the Permeate turbidity and DOC, however, the claim is not reliable because there was no statistical analysis was presented (error bar).
  8. Extensive editing of English language and style required. For example, title is not clear, and can be improved.

Author Response

Comment 1:In abstract, please remove the words “Background”; “Methods”, “Results”, and “Conclusions”

Re: Thanks for the comment. We removed these words in the abstract of revised manuscript.

Comment 2:What is the difference of this work from the other reported literature, like Water research, 41(8), 1713-1722; Journal of Membrane Science 281, no. 1-2 (2006): 716-725, Journal of Environmental Sciences, 18(5), 880-884.

Re: Thanks for the comment. The mentioned papers are as follows:

[1] Jermann D, Pronk W, Meylan S, et al. Interplay of different NOM fouling mechanisms during ultrafiltration for drinking water production[J]. Water Research, 2007, 41(8):1713-1722.

[2] Costa A R, Pinho M N D, Elimelech M. Mechanisms of colloidal natural organic matter fouling in ultrafiltration[J]. Journal of Membrane Science, 2006, 281(1-2):716-725.

[3] Jin W, Xiao-Chang W. Ultrafiltration with in-line coagulation for the removal of natural humic acid and membrane fouling mechanism[J]. Journal of Environmental Sciences, 2006, 18(5):880-884.

UF membrane (PES, 100 kDa) with dead-end filtration mode was used in reference [1]. The test water was synthetic lake water using model natural organic matters (NOMs). The impact of molecular interactions between different NOMs on UF fouling mechanisms was investigated in detail. The results showed that polysaccharide was one of the main UF foulants and calcium affected the interaction between NOMs.

Two laboratory-made cellulose acetate UF membranes with dead-end filtration mode and humic acid model solution were used in reference [2]. Membrane fouling mechanism analysis showed that at the earlier stage of filtration, the dominant fouling mechanism was pore blocking for both UF membranes.

Hollow fiber UF membrane (PAN, 100 kDa), humic acid model solution and in-line coagulation pre-treatment were used in reference [3]. The results showed that in-line coagulation reduced the rate of membrane fouling and resulted in more constant permeate flux and very slight increase of transmembrane pressure during a filtration circle. Membrane fouling by both pore-narrowing effect and cake layer was effectively mitigated by pre-coagulation.

Reference [ 1], [2], and [3] focused on UF membrane fouling by model NOMs (mainly humic acid) solution from the perspective of physicochemical interactions among NOMs, UF membrane and matrix (e.g., calcium). Our study focused on UF membrane fouling by real raw and coagulated lake water from the perspective of mathematical model fitting including the established single and combined fouling models. Thus, our study is significantly different from the above-mentioned references. We emphasized this point in Line 73-75 in the revised manuscript.

Comment 3:In introduction, it is important to mention how ultrafiltration used for production of drinking water. Is it a pretreatment for reverse osmosis or distillation process to obtain drinkable water? How it is a help in the overall process for production of clean water?

Re: Thanks for the comment. There are mainly two ways for UF for drinking water production. One is the drinking water production from conventional surface water resources (e.g., river, lake, reservoir), where UF either serves as advanced treatment after conventional treatment (i.e., coagulation – sedimentation - media filtration) – the dominant application in China, or replaces the sedimentation and media filtration in conventional treatment to form the so-called “coagulation – ultrafiltration” process. The other is the drinking water production from unconventional saline water resources (e.g., brackish and sea water), where UF mainly serves as the pre-treatment prior to reverse osmosis desalination. As far as we know, there are two large-size UF drinking water treatment plants commissioned in this year in China. One is pressurized UF system with capacity of 0.6 million m3/d in Guangzhou, the other is submerged UF system with capacity of 0.5 million m3/d in Ningbo. The increasing demand for high quality drinking water and the decreasing cost of UF technology would drive the UF application for drinking water production, we think. We briefly added this point in Line 30-34 and Line 38-41 in the revised manuscript.

Comment 4:The surface water was gathered in the lake, does the turbidity and dissolved organic carbon is similar if gathered in a different day?

Re: Thanks for the comment. We had taken samples at the same place in different days and found that the turbidity was mainly affected by rain and went up to over 100 NTU just after raining.

Comment 5:Is the fouling mechanism is similar to the authors finding, if the property of the surface water changed?

Re: Thanks for the comment. The fouling mechanism is intrinsically dependent on the interactions between surface water and UF membrane. If the property of the surface water (e.g., particle size distribution) changed while UF membrane unchanged, the fouling mechanism would also change.

Comment 6: If the MWCO of the UF membrane was changed, is the fouling mechanism is similar? Why the author chose PVDF UF membrane with MWCO of 150 kDa? MWCO of UF membrane had a large range. It is interesting if the author could include this in their study.

Re: Thanks for the comment. Following the previous comment, if MWCO of the UF membrane changed while surface water unchanged, the fouling mechanism would also change. The main idea of this study is to clarify the UF membrane fouling mechanisms via mathematical model fitting including the established single and combined fouling models. This methodology would be universal for any UF membrane. Thus, the selection of UF membrane is mainly based on our lab availability. The MWCO (or pore size) of UF membrane for drinking water treatment is generally compromised between rejection capability and permeability. As far as we know, there seems a consensus that the nominal pore size of 10-20 nm is reasonable for UF membrane used for drinking water production in order to meet virus rejection requirements and keep the high permeability. We added this point in Line 30-34 and Line 274-277 in the revised manuscript.

Comment 7: In figure 2, the author mentioned that there was no significant change on the Permeate turbidity and DOC, however, the claim is not reliable because there was no statistical analysis was presented (error bar).

Re: Thanks for the comment. We put the error bar on Figure 2 and revised the data analysis in Line 141 in the revised manuscript.

Comment 8: Extensive editing of English language and style required. For example, title is not clear, and can be improved.

Re: Thanks for the comment. Following the comments of you and other reviewers, we carefully corrected all editing and style mistakes in red in the revised manuscript. The title was changed into “Fouling Mechanisms Analysis via Combined Fouling Models for Surface Water Ultrafiltration Process” for better understanding.

Reviewer 4 Report

Article entitled Ultrafiltration Fouling Mechanisms for Surface Water Treatment Based on Combined Fouling Models written by Bin Huang, Hangkun Gu, Kang Xiao, Fangshu Qu, Huarong Yu, Chunhai Wei and submitted to Membranes journal as a draft no 859906, deals with an important issue of fouling mechanisms modeling.

The article is interesting and could be considered for publication in Membranes journal. As English is not my native language, I am not able to assess its correctness. However, while reading, I found some statements missing, confusing or unclear. Below I enclose a list of my comments.

I suggest transfer lines 84 (starting from: Turbidity removal) – 93 to Results part, as it was a part of experiment. What is more, I suggest connecting 2.1 and 2.4 in one paragraph. Lines 120 – 123 just after line 79.

Was coagulation performed in natural pH of water (any value is known?) or was there any pH adjustment?

Is there any chance to increase the quality of Fig 4 – pixels are visible with a naked eye.

Lines 44 – 74, 120 – 132, 155 – 182, 218 – 237, 243 – 277, 279 - 316 – text editing.

Based on my comments and general impression, I suggest to accept this article but after some small changes.

Author Response

The article is interesting and could be considered for publication in Membranes journal. As English is not my native language, I am not able to assess its correctness. However, while reading, I found some statements missing, confusing or unclear. Below I enclose a list of my comments.

Comment 1: I suggest transfer lines 84 (starting from: Turbidity removal) – 93 to Results part, as it was a part of experiment. What is more, I suggest connecting 2.1 and 2.4 in one paragraph. Lines 120 – 123 just after line 79.

Re:Thanks for the comment. Our study focused on UF fouling mechanism analysis via mathematical model fitting including single and combined fouling models. Coagulation tests were aimed to produce coagulated surface water to simulate the hybrid coagulation – UF process. Thus, it is better to briefly show the data in the section of 2.2 Coagulation Pretreatment rather than in the main section of 3. Results and Discussion. Section 2.1 to 2.3 follows the experimental sequence. It is better to remain it unchanged, we think. Hope you agree with us.

Comment 2: Was coagulation performed in natural pH of water (any value is known?) or was there any pH adjustment?

Re:Thanks for the comment. Coagulation was performed in natural pH (7.34) of lake water. We added pH information in Line 87-88 in the revised manuscript.

Comment 3: Is there any chance to increase the quality of Fig 4 – pixels are visible with a naked eye.

Re:Thanks for the comment. We reproduced the Fig 4 in the revised manuscript.

Comment 4: Lines 44 – 74, 120 – 132, 155 – 182, 218 – 237, 243 – 277, 279 - 316 – text editing.

Re:Thanks for the comment. Following the comments of you and other reviewers, we carefully corrected all editing mistakes in red in the revised manuscript.

Round 2

Reviewer 3 Report

The authors carefully addressed my concerns. Thus, the manuscript is acceptable for publication in Membranes.